# Peptides Are Cardioprotective Drugs of the Future: The Receptor and Signaling Mechanisms of the Cardioprotective Effect of Glucagon-like Peptide-1 Receptor Agonists

**DOI:** 10.3390/ijms25094900

**Published:** 2024-04-30

**Authors:** Alla A. Boshchenko, Leonid N. Maslov, Alexander V. Mukhomedzyanov, Olga A. Zhuravleva, Alisa S. Slidnevskaya, Natalia V. Naryzhnaya, Arina S. Zinovieva, Philipp A. Ilinykh

**Affiliations:** 1Department of Atherosclerosis and Chronic Coronary Heart Disease, Cardiology Research Institute, Tomsk National Research Medical Center, Russian Academy of Sciences, 634012 Tomsk, Russia; 2Laboratory of Experimental Cardiology, Cardiology Research Institute, Tomsk National Research Medical Center, Russian Academy of Sciences, 634012 Tomsk, Russia; 3Department of Pathology, The University of Texas Medical Branch at Galveston, Galveston, TX 77555, USA

**Keywords:** glucagon-like peptide-1, regulated cell death, heart, ischemia/reperfusion, hypertension, diabetic cardiomyopathy, adverse remodeling of the heart, kinases

## Abstract

The high mortality rate among patients with acute myocardial infarction (AMI) is one of the main problems of modern cardiology. It is quite obvious that there is an urgent need to create more effective drugs for the treatment of AMI than those currently used in the clinic. Such drugs could be enzyme-resistant peptide analogs of glucagon-like peptide-1 (GLP-1). GLP-1 receptor (GLP1R) agonists can prevent ischemia/reperfusion (I/R) cardiac injury. In addition, chronic administration of GLP1R agonists can alleviate the development of adverse cardiac remodeling in myocardial infarction, hypertension, and diabetes mellitus. GLP1R agonists can protect the heart against oxidative stress and reduce proinflammatory cytokine (IL-1β, TNF-α, IL-6, and MCP-1) expression in the myocardium. GLP1R stimulation inhibits apoptosis, necroptosis, pyroptosis, and ferroptosis of cardiomyocytes. The activation of the GLP1R augments autophagy and mitophagy in the myocardium. GLP1R agonists downregulate reactive species generation through the activation of Epac and the GLP1R/PI3K/Akt/survivin pathway. The GLP1R, kinases (PKCε, PKA, Akt, AMPK, PI3K, ERK1/2, mTOR, GSK-3β, PKG, MEK1/2, and MKK3), enzymes (HO-1 and eNOS), transcription factors (STAT3, CREB, Nrf2, and FoxO3), K_ATP_ channel opening, and MPT pore closing are involved in the cardioprotective effect of GLP1R agonists.

## 1. Introduction

Despite the achievements of modern cardiology, mortality in patients with acute myocardial infarction (AMI) remains high [1,2,3,4,5]. Mortality is especially high in patients with large infarct size, microvascular obstruction, and cardiogenic shock [6,7,8]. A related problem is postinfarction heart remodeling and heart failure (HF). In recent years, there has been no noticeable decrease in mortality in patients with AMI. It is quite obvious that there is an urgent need to develop new, more effective drugs for the treatment of AMI and HF. Of particular interest in this regard is glucagon-like peptide-1 (GLP-1). This hormone enters the blood from α-cells of the islets of Langerhans and L-cells of the small intestine [9]. It was found in neurons of the nucleus of the solitary tract [9]. The main physiological function of GLP-1 is the regulation of glucose metabolism. GLP-1 increases insulin secretion and reduces glucagon secretion [9]. GLP1 receptor (GLP1R) agonists reduced appetite and induced weight loss in humans with obesity [10]. It is believed that a decrease in appetite occurs due to the activation of GLP-1 receptors (GLP1Rs) localized in the brain and the intestine, decelerating a process of gastric emptying [10,11]. Intracerebroventricular administration of GLP-1 reduced feeding in rats [12].

The GLP1R is a G protein-coupled receptor [9]. It can interact with G_s_, G_i/o_, G_q_, and G_11_ proteins [9,13,14,15,16,17]. It should be noted that G_i/o_, G_q_, and G_11_ proteins are involved in the cardioprotective effect of cannabinoids, opioids, adenosine, and bradykinin [18,19,20]. GLP1R is subjected to internalization after the interaction with β-arrestin [21,22].

GLP-1 and glucagon are synthesized from preproglucagon by enzymatic cleavage [9]. GLP-1 circulating in blood is represented by two peptides: GLP-1(7–36) and GLP-1(7–37) (Table 1) [23]. These peptides are selective agonists of the GLP1R [23]. These peptides are quickly cleavaged in blood and tissues by dipeptidyl peptidase 4 (DPP-4) and neutral endopeptidase [11]. The half-life of GLP-1 is less than 2 min [9,11], which does not allow its use as a drug. Enzyme-resistant analogs are needed.

In the last 19 years, information has appeared suggesting that GLP-1 and its enzyme-resistant analogs can increase the heart’s resistance to ischemia/reperfusion (I/R) [24,25,26,27,28,29,30,31,32,33,34,35,36,37,38,39,40,41,42,43,44,45,46,47,48,49,50].

The objective of this review is an analysis of data on the molecular mechanism of the cardioprotective effects of glucagon-like peptide-1 and its enzyme-resistant analogs.

## 2. Discovery of Glucagon-like Peptide-1 and Exendin

In 1979 and 1980, it was discovered that, in addition to glucagon, there is glucagon-like peptide-1 [51,52,53]. In 1989, two teams of investigators identified the molecular structure of GLP-1(7–37) [13,54]. In 1996, it was found that glucagon and GLP-1 are formed from the high molecular weight precursor proglucagon which contains 178 amino acid residues [55].

As we have already mentioned above, GLP-1 is highly sensitive to enzymatic hydrolysis, so it is difficult to study its effects in vivo. Progress began with the discovery of its enzyme-resistant analog exendin [14,56]. In 1990, polypeptide exendin-3 was isolated from the venom of the Gila monster lizard, *Heloderma horridum* [56]. Exendin-3 contains 39 amino acid residues (Table 2). It has the greatest homology with human GLP-1 (50%) [56]. In 1992, the polypeptide exendin-4, which contains 39 amino acid residues (Table 2), was isolated from the venom of lizard, *Heloderma suspectum* [14]. It was found that exendin-4 is a more effective GLP-1 agonist than exendin-3 [14]. This polypeptide is resistant to enzymatic hydrolysis by DPP-4 and neutral aminopeptidase [11]. In 1991, it was found to be a selective peptide GLP1R antagonist [57]. It was truncated exendin-4 (9–39) (Table 2) [57]. The ability of exendin-4 (9–39) to block the effects of GLP1 was confirmed by Göke et al. [58].

Exendin-4 stimulates β-cell proliferation and regeneration of islets of Langerhans [59]. The polypeptide decreased fasting and postprandial glucose and reduced energy intake in healthy volunteers [60]. Exendin-4 reduced the plasma glucose level and decreased insulin resistance in hyperglycemic db/db and ob/ob mice [61]. Exendin-4 dose-dependently induced glucose lowering in diabetic rhesus monkeys and diabetic fatty Zucker rats [61]. Consequently, exendin-4 could be used for the treatment of type 2 diabetes.

## 3. Permeability of the Blood–Brain Barrier of Glucagon-like Peptide-1 and Its Analogs

Could GLP-1 and its analogs cross the blood–brain barrier (BBB) and exhibit the central effects? Small hydrophobic molecules easily cross the BBB [62,63]. In contrast, the BBB is a serious barrier to the penetration of large hydrophilic molecules [62,63]. For example, hydrophilic opioid peptide D-Ala^2^,D-Leu^5^-enkephalin (dalargin) can cross the BBB only at a dose of 0.5 mg/kg [64]. GLP-1 is a large hydrophilic molecule. Exogenous GLP-1 will be unable to cross the BBB because proteases rapidly cleavage it in blood. However, large hydrophilic molecules also can penetrate into the brain if they are resistant to enzymatic hydrolysis; for example, erythropoietin can cross the BBB [65].

These findings demonstrate that enzyme-resistant GLP-1 analogs could cross the BBB. The half-life of exendin-4 is 2.4–4 h, allowing it to penetrate into the brain after intravenous administration [66,67]. It was found that exendin-4 is soluble in water but exhibits moderate lipophilicity, allowing it to quickly penetrate into the brain after intravenous injection regardless of circumventricular organs which are devoid of the BBB [68]. The GLP1R antagonist exendin (9–39) can apparently cross the BBB. Indeed, it was found that intravenous infusion of exendin (9–39) with the rate of 30 nmol/kg/min (101 µg/kg/min) for 20 min resulted in a blockade of the central GLP1R (a total dose is 2.02 mg/kg) [69]. Exendin (9–39) eliminated the cardioprotective effect of exendin-4 at a dose of 5 μg/kg [50]. It could be suggested that exendin (9–39) blocks only the peripheral GLP1R at this dose. However, exendin (9–39) at a dose of 50 μg/kg abolished the neuroprotective effect of remote conditioning in rats with middle cerebral artery occlusion (90 min) and reperfusion (24 h) [70]. Consequently, endogenous GLP1 can protect the brain against I/R. Exendin (9–39) can penetrate into the brain at a dose of 50 μg/kg.

The peptide GLP1R agonist albiglutide has a half-life ranging between 6 and 7 days in humans [71]. ^125^I-labeled peptide GLP1R agonists albiglutide, dulaglutide, and tirzepatide penetrated into the brain of mice after intravenous administration [72]. Albiglutide and dulaglutide crossed the BBB quickly within 1 h after injection. Tirzepatide penetrated into the brain only 6 h after injection [72]. The peptide GLP1R agonist liraglutide penetrated into the brain in humans [73]. ^125^I-labeled peptide GLP1R agonists exendin-4, liraglutide, lixisenatide, and semaglutide crossed the BBB in mice but with different rates [74]. Exendin-4 penetrated into the brain most quickly. The exendin-4 brain/serum ratio was about 25% 50 min after intravenous injection [74]. The exendin-4 brain/serum ratio was about 50% 100 min after intravenous injection. Liraglutide and semaglutide crossed the BBB most slowly. The liraglutide and semaglutide brain/serum ratio was less than 1% 100 min after intravenous injection [74]. These polypeptides apparently penetrate into the brain but very slowly. The half-life of liraglutide is 12.6 ± 1.1 h in healthy volunteers [75]. Consequently, liraglutide is resistant to enzymatic hydrolysis and, when administered daily, it can occupy central GLP1Rs and exhibit a neuroprotective effect [76]. Chronic administration of this polypeptide to patients with type 2 diabetes resulted in the appearance of liraglutide in cerebrospinal fluid [73]. It was found that the anti-obesity effect of liraglutide is a result of the activation of hypothalamic neurons [77]. The plasma half-life of semaglutide (ozempic) is 46.1 h in mini-pigs after intravenous administration [78]. Both semaglutide and liraglutide exhibited neuroprotective properties in rats with middle cerebral artery occlusion (90, 120, or 180 min) and reperfusion (24 h) [79].

Thus, all peptide GLP1R agonists can cross the BBB. Exendin-4 occupies central GLP1R 1 h after intravenous administration. Semaglutide and liraglutide slowly penetrate into the brain. It requires some hours for them to penetrate into the brain.

## 4. The Infarct-Reducing Effect of GLP1R Agonists

Coronary artery occlusion (CAO, 30 min) and reperfusion (120 min) were performed in rats [24]. The cardioprotective effect of GLP-1 was studied both in vivo and in vitro. The infarct size/area at risk (IS/AAR) ratio was assessed. GLP-1 was infused intravenously at a rate of 4.8 pmol/kg/min throughout the I/R period. GLP-1 reduced the IS/AAR ratio by 42%. The isolated rat heart was subjected to CAO (35 min) and reperfusion [24]. Hearts were perfused with a solution containing GLP-1 at a final concentration of 3 nmol/L for 5 min before CAO. GLP-1 reduced the IS/AAR ratio by 45%. The selective GLP1R antagonist exendin (9–39) eliminated the cardioprotective effect of GLP-1 [24]. Therefore, the infarct-reducing effect of GLP-1 is associated with the activation of the cardiac GLP1R. Isolated rat hearts were subjected to global ischemia (45 min) and reperfusion (120 min) [36]. The selective GLP1R agonist exendin-4 was added to the perfusion solution at the onset of reperfusion. The duration of perfusion with a solution containing exendin-4 was 15 min. Exendin-4 reduced infarct size by about 50%. In addition, exendin-4 contributed to the improved recovery of cardiac contractile function in reperfusion [36]. Mice were subjected to permanent CAO (without reperfusion) [44]. The selective peptide GLP1R agonist liraglutide was administered to animals for 7 days. The drug increased the survival of mice with myocardial infarction (MI). Liraglutide reduced infarct size by 20% (n = 36) 4 days after CAO [44]. Preliminary perfusion of isolated mouse hearts with a solution containing liraglutide increased cardiac tolerance to I/R. The same effect was achieved by the use of liraglutide in reperfusion [44].

The isolated rat heart was subjected to global I/R [45]. In reperfusion, the heart was perfused for 15 min with a solution containing GLP-1 at a final concentration of 0.3 nmol/L. GLP-1 reduced infarct size by 39%. The selective GLP1R antagonist exendin (9–39) eliminated this effect [45]. These data indicate that GLP-1 increases the tolerance of the heart to reperfusion due to the activation of the cardiac GLP1R. The isolated mouse heart was subjected to global ischemia (30 min) and reperfusion (120 min) [46]. The heart was reperfused with a solution containing GLP-1(9–36)amide (0.3 nmol/L) or the enzyme-resistant peptide GLP1R agonist exendin-4 (3 nmol/L). GLP-1(9–36) amide reduced infarct size by 35%, exendin-4 reduced infarct size by 53%. Both peptides improved recovery of cardiac contractility in reperfusion [46]. Peptides increased the tolerance of cardiomyocytes to hypoxia/reoxygenation (H/R). The GLP1R antagonist exendin (9–39) eliminated the cytoprotective effect of these peptides [46]. These data suggest that the infarct-limiting effect of peptide GLP1R agonists is associated with the activation of the GLP1R in cardiomyocytes.

Coronary artery ligation (30 min) and reperfusion (180 min) were carried out in rabbits [80]. GLP-1-Tf, a peptide GLP1R agonist, was administered at a dose of 10 mg/kg before ischemia or before reperfusion. In both cases, peptide reduced infarct size by 25% [80]. Rats were subjected to CAO (30 min) and reperfusion (24 h) [47]. Albiglutide, an enzyme-resistant peptide GLP1R agonist, was administered subcutaneously daily for three days before CAO. This peptide reduced infarct size by 26% at a dose of 3 mg/kg/day. Further, an increase in the dose did not augment the infarct-limiting effect. Peptide did not affect cardiac contractility [47]. CAO (30 min) and reperfusion (120 min) were induced in rats [48]. GLP-1 was infused intravenously (30 pmol/kg/min) throughout the reperfusion period, beginning 2 min before the restoration of coronary blood flow. It was found that GLP-1 reduced infarct size by 79% (n = 5). According to our data, to objectively assess infarct size, there must be at least 8 animals in the group, so such a pronounced infarct-reducing effect of GLP-1 could be the result of a statistical error. In addition, it was found that GLP-1 reduced neutrophil invasion into the infarcted myocardium, which could also contribute to the reduction in reperfusion injury of the heart [48]. Isolated rat hearts were subjected to global ischemia (35 min) and reperfusion (120 min) [49]. In reperfusion, hearts were perfused for 15 min with a solution containing peptide GLP1R agonists exendin-4 or curaglutide (N-Ac-GLP-1(7–34)amide). Curaglutide reduced infarct size by 64%, and exendin-4 reduced infarct size by 49%. Both peptides improved cardiac contractility in reperfusion. The GLP1R antagonist exendin (9–39) eliminated the infarct-sparing effect of curaglutide [49]. Consequently, stimulation of the cardiac GLP1R could reduce cardiac reperfusion injury.

Coronary artery occlusion (30 min) and reperfusion (4 h) were performed in rats [50]. Exendin-4 (5 μg/kg) was injected intravenously 1 h before ischemia. This peptide reduced infarct size by approximately 50%. The GLP1R antagonist exendin (9–39) (5 μg/kg intravenously) eliminated the cardioprotective effect of exendin-4 [50]. Consequently, stimulation of the GLP1R increases cardiac resistance to I/R. Coronary artery occlusion (45 min) and reperfusion (120 min) were performed in the isolated rat heart [26]. The heart was perfused with a solution containing the peptide GLP1R agonist lixisenatide (0.3 nmol/L) 10 min before the onset of reperfusion. Lixisenatide reduced infarct size by 25% but did not affect myocardial contractility. In vivo, CAO (30 min) and reperfusion were carried out in rats [26]. The next day, a 10-week-course administration of lixisenatide (10 µg/kg/day subcutaneously) was initiated. The drug reduced left ventricular end-diastolic pressure (LVEDP) by 57% compared to the control group (I/R without drugs). The brain natriuretic peptide (BNP) level was reduced by 39% [26]. Consequently, lixisenatide partially prevented the development of heart failure. However, the peptide had no effect on post-infarction cardiac fibrosis. Rats were subjected to CAO (30 min) and reperfusion (6 h) [27]. Exendin-4 (1 μg/kg) was administered 30 min before ischemia. The peptide reduced infarct size by 22.5%. Increasing the dose to 10 µg/kg did not lead to an increase in the infarct-limiting effect [27].

Coronary artery ligation (30 min) and reperfusion (120 min) were carried out in rats [81]. Exendin-4 was administered at a dose of 10 μg/kg/day intraperitoneally for 10 days before MI. It was found that preliminary administration of this peptide contributed to a reduction in infarct size by 30% [82]. Injection of exendin (1 μg/kg, subcutaneously) 1 h before CAO reduced infarct size by 49%. Increasing the dose of peptide (5, 10 μg/kg) did not lead to an increase in the infarct-sparing effect [82]. Mice were subjected to CAO (30 min) and reperfusion [28]. Exendin was administered subcutaneously 60 min before CAO. Peptide at a dose of 1 μg/kg caused a reduction in infarct size of 49%. An increase in the dose to 2.5 and 5 μg/kg did not augment the infarct-limiting effect of exendin [28].

Isolated rat cardiomyocytes were exposed to anoxia (1 h) and reoxygenation (1 h) [29]. Exendin-4 (3 nmol/L) increased the number of surviving cardiomyocytes. Methyl-β-cyclodextrin, an inhibitor of caveolae formation, eliminated the cytoprotective effect of exendin-4. Exendin-4 increased the expression of caveolin-3 protein, which is involved in the formation of caveolae [29]. The GLP1R was co-localized with caveolin-3. Coronary artery occlusion (30 min) and reperfusion (2 h) were carried out in mice [29]. Exendin-4 caused a 40% reduction in infarct size. Exendin-4 did not affect infarct size in mice with caveolin-3 knockout [29]. The important role of caveolin-3 and caveolae in the cardioprotective effect of GLP1R agonists was confirmed by Hamaguchi et al. [30]. Coronary artery occlusion (30 min) and reperfusion (2 h) were induced in mice. Exendin-4 was administered intravenously before ischemia at a dose of 3 or 30 ng/kg. Polypeptide at a dose of 30 ng/kg reduced infarct size by 44% and had no effect on infarct size at a dose of 3 ng/kg. Exendin-4 did not have a cardioprotective effect in mice with caveolin-3 knockout [30]. Therefore, the presence of caveolin-3 and caveolae is required for normal GLP1R function.

Coronary artery ligation (60 min) and reperfusion (180 min) were carried out in mice [34]. The GLP1R agonist liraglutide was administered at a dose of 10 μg/kg subcutaneously for 5 days before I/R. Liraglutide reduced infarct size by 40% and improved the contractile function of the heart when coronary blood flow was restored [34]. Liraglutide reduced serum levels of the proinflammatory cytokines interleukin-6 (IL-6) and tumor necrosis factor-α (TNF-α). This GLP1R agonist reduced the levels of pro-inflammatory cytokines in myocardial tissue: IL-6 and monocyte chemoattractant protein-1 (MCP-1). In addition, the peptide reduced collagen-3 expression in the infarcted myocardium [34]. The DPP-4 inhibitor linagliptin did not augment the infarct-reducing effect of liraglutide [34]. Coronary artery occlusion (45 min) and reperfusion (75 or 120 min) were performed in pigs [38]. Liraglutide (4.8 mg/mL) was administered intravenously at a rate of 6 mL/min, beginning 15 min after the onset of ischemia and throughout reperfusion. Postconditioning was carried out using four sessions of CAO (30 s) and reperfusion (30 s) after 45 min of CAO. Investigators were unable to detect the cardioprotective effect of liraglutide and postconditioning [38]. Apparently, there was an error in the experimental technique, since the phenomenon of postconditioning was well reproduced in many studies [83].

Coronary artery ligation (30 min) and reperfusion (4 h) were performed in rats [41]. Exendin-4 (140 ng/kg) was administered intravenously before CAO or 10 min after coronary artery ligation. Exendin reduced infarct size by about 90%, while the level of myocardial necrosis markers creatine kinase-MB (CK-MB) and troponin I in serum decreased by only 60% [41]. In addition, exendin-4 improved contractility in reperfusion. Exendin-4 increased the ATP level in the infarcted myocardium, which could promote the restoration of cardiac contractile function. Thus, there is a significant discrepancy in quantitative estimates of I/R damage to the heart. It is possible that a subjective element played a role in the morphological assessment of infarct size. The small dose of exendin-4 (140 ng/kg) was also surprising. Many of the above-mentioned studies used exendin-4 at a dose of 1–5 μg/kg [27,82,84]. An exception is the work of Hamaguchi et al., who discovered the infarct-limiting effect of exendin at a dose of 30 ng/kg [30].

Isolated rat hearts were exposed to regional ischemia (30 min) and reperfusion (30 min) [42]. The hearts were perfused with a solution containing exendin-4 (100 nmol/L) or GLP-1(9–36) (100 nmol/L) 5 min before reperfusion and perfusion was continued for 10 min of reperfusion. Exendin-4 and GLP-1(9–36) improved cardiac contractility in reperfusion. The GLP1R antagonist exendin (9–39) (100 nmol/L) abolished the inotropic effect of exendin-4 but did not affect the inotropic effect of GLP-1(9–36). Exendin-4 and GLP-1(9–36) reduced the reperfusion creatine kinase release from the myocardium. The GLP1R antagonist exendin (9–39) eliminated the cardioprotective effect of exendin-4 but did not affect the cardioprotective effect of GLP-1(9–36) [42]. It was concluded that the protective effect of exendin-4 is accompanied by the activation of the GLP1R, and the protective effect of GLP-1(9–36) does not depend on stimulation of this receptor [42]. CAO (30 min) and reperfusion were carried out in rats [43]. The GLP1R agonist liraglutide was administered daily for 7 days before I/R, beginning at a dose of 35 μg/kg/day. Peptide reduced infarct size, decreased the serum CK-MB level, and improved contractility of the myocardium. The cardioprotective effect of liraglutide reached a maximum at a dose of 140 µg/kg/day [43]. H9c2 cardiomyoblasts were subjected to hypoxia (4 h) and reoxygenation (4 h) with and without liraglutide (25, 50, 100, 200, or 400 nmol/L) [43]. Liraglutide increased cell survival and decreased the release of lactate dehydrogenase, a marker of sarcolemmal injury, from cells. The cytoprotective effect reached a maximum at a concentration of 400 nmol/L. These findings indicate that the cardioprotective effect of liraglutide in vivo appears to result from the activation of the GLP1R in cardiomyocytes.

Thus, it was convincingly shown that stimulation of the GLP1R before I/R leads to the limitations of infarct size and the improvement of contractility in reperfusion. The effect of GLP-1 agonists on I/R seems to be typical for all classes of drugs. The activation of the GLP1R in reperfusion also reduces infarct size and improves postischemic recovery of contractility. The cardioprotective effect of GLP1R agonists is associated with the activation of receptors localized in the myocardium and cardiomyocytes. However, we cannot exclude the involvement of other extracardiac mechanisms in the cardioprotective effect of GLP1R agonists in vivo. The infarct-reducing effect of exendin-4 in vivo could be mediated via stimulation of central GLP1Rs because it crosses the BBB.

## 5. Glucagon-like Peptide-1 and Regulated Forms of Cell Death

There are five main forms of regulated cardiomyocyte death: apoptosis, necroptosis, pyroptosis, autophagy, and ferroptosis [85,86,87,88,89]. There is a sixth form of cell death: paraptosis [90]. However, its role in cardiac injury has not yet been proven.

Chronic subcutaneous infusion (3 months) of GLP-1 inhibits apoptosis of cardiomyocytes in spontaneously hypertensive, heart failure-prone (SHHF) rats [91]. GLP-1 inhibited apoptosis of isolated rat cardiomyocytes subjected to H/R [92]. Chronic pretreatment with the DPP-4 blocker sitagliptin protected the rat heart against I/R and inhibited apoptosis in the myocardium [82]. Exendin-4 increased the survival of H9c2 cells and inhibited apoptosis in H/R [93]. In addition, exendin-4 reversed mitochondrial permeability transition pore (MPT pore) opening and reduced the active caspase-3 level in cells [93]. The high glucose concentration induced apoptosis of H9c2 cells [94]. Liraglutide reversed apoptosis, inhibited reactive oxygen species (ROS) production, increased the expression of exchange protein activated by cAMP-1 (Epac-1), an intracellular sensor of cAMP, and activated Akt [94]. Oxidative stress in isolated neonatal rat cardiomyocytes was induced by incubation with H_2_O_2_ [95]. Exendin-4 decreased ROS production, inhibited apoptosis and caspase-3 activity, upregulated superoxide dismutase (SOD), catalase, glutathione peroxidase-1 (GPx-1) activity, and increased the antiapoptotic protein Bcl2 level in cells. The exendin-triggered antiapoptotic effect was mediated through cAMP-dependent protein kinase A (PKA) and Epac [95].

Streptozotocin-induced diabetic rats were treated with liraglutide for 4 weeks [96]. Liraglutide inhibited apoptosis of cardiomyocytes in diabetic hearts [96]. H9c2 cells were exposed to H/R [97]. Exendin-4 increased cell viability and inhibited apoptosis [97]. Exendin-4 inhibited apoptosis of cardiomyocytes in rats with streptozotocin-induced diabetes [98]. Liraglutide inhibited apoptosis in the myocardium in rats with streptozotocin-induced diabetes [99].

Apoptosis of isolated neonatal rat cardiomyocytes was induced by TNF-α [100]. Exendin-4 inhibited ROS production and apoptosis of rat cardiomyocytes [100]. H9c2 cells and neonatal rat ventricular cardiomyocytes were exposed to H/R [101]. Liraglutide alleviated apoptosis of H9c2 cells and cardiomyocytes, inhibited ROS production, and reduced mitochondrial injury. Liraglutide mitigated H/R-induced Ca^2+^ overload of H9c2 cells. H/R decreased calcium transient in cardiomyocytes. Liraglutide augmented the expression of sarcoplasmic reticulum Ca^2+^-ATPase and increased calcium transient [100]. However, the investigators used small groups (n = 3) [100]. Therefore, the significance of these findings is questionable.

Diabetic mice db/db underwent CAO (30 min) and reperfusion (3 h) [102]. The peptide GLP1R and glucagon receptor agonist ZP2495 improved contractility of the heart and inhibited cardiomyocyte apoptosis. Neonatal rat ventricular cardiomyocytes were subjected to H/R. ZP2495 blocked MPT pore opening and mitochondrial ROS generation. The cytoprotective effect of the GLP1R agonist was associated with an increase in the activated p-Akt kinase, phosphorylated AMP-activated protein kinase (p-AMPK), and antiapoptotic Bcl2 levels [102]. ZP2495 improved mitochondrial respiration, increased the ATP level, and the expression of the transcription factor Forkhead box O3 (FoxO3) in the infarcted myocardium [102].

H9c2 cells were exposed to H/R [103]. GLP-1 increased cell viability, and reduced lactate dehydrogenase and CK-MB release. GLP-1 alleviated H/R-induced apoptosis of H9c2 cells, downregulated proapoptotic protein Bax content, increased the antiapoptotic protein Bcl2 level, and reduced caspase-3 activity. GLP-1 increased the p-Akt level. GLP-1 reduced the H/R-induced expression of endoplasmic reticulum stress proteins GRP78, CHOP, and caspase-12, where CHOP is a C/EBP homologous protein, and GRP78 is a 78 kDa glucose-regulated protein. These effects were completely abolished by exendin (9–39) and phosphoinositide 3-kinase (PI3K) inhibitor LY294002 [103]. Consequently, the GLP1R and the PI3K/Akt pathway are involved in the protective effect of GLP-1. Hypoxia/reoxygenation induced apoptosis of H9c2 cells [104]. Liraglutide reduced apoptosis and increased cell viability. Polypeptide upregulated Bcl2 expression and downregulated Bax expression. Liraglutide increased the expression of Notch1 and Jagged1, where Notch1 is Notch homolog 1, a translocation-associated (Drosophila) pro-survival protein preventing cardiomyocyte apoptosis through the activation of the PI3K/Akt pathway [104]. Jagged1 is a Notch1 ligand [104]. Investigators concluded that the Notch pathway is responsible for the antiapoptotic effect of liraglutide [104].

Rats underwent CAO (30 min) and reperfusion (2 h) [105]. Liraglutide was administered at a dose of 0.18 mg/kg 12 h before CAO. The GLP1R agonist reduced infarct size by 34% and decreased the number of apoptotic cells by 43%. The heat shock protein-90 (HSP90) inhibitor geldanamycin (1 mg/kg) abolished both of the protective effects of polypeptide [105]. Consequently, HSP90 is involved in the infarct-reducing and antiapoptotic effects of GLP1R stimulation. Rats were subjected to coronary artery ligation and reperfusion (4 h) [106]. Semaglutide was injected into rats 30 min before I/R. Polypeptide upregulated GLP1R expression, increased the levels of phosphorylated protein kinase G (p-PKG), phosphorylated extracellular signal-regulated kinase (p-ERK), and phosphorylated protein kinase C-ε (p-PKCε). Semaglutide reduced infarct size by about 70% and inhibited apoptosis. The GLP1R agonist upregulated Bcl2 and downregulated cleaved caspase-3, Bax [106]. The PKG inhibitor KT-5823 partially abolished these effects of semaglutide. 8-Br-cGMP, a cGMP analog, mimics the cardioprotective effect of polypeptide. Investigators concluded the cardioprotective effect of GLP1R stimulation is mediated through the activation of the PKG/PKCε/ERK1/2 pathway [106].

These data convincingly demonstrate that GLP1R stimulation inhibits the development of myocardial apoptosis in diabetic and nondiabetic rats through the activation of the PKG/PKCε/ERK1/2 pathway, Akt, and AMPK, the upregulation of HSP90 and Bcl2, and the downregulation of GRP78, CHOP, and Bax. 

Injection of exendin (1 μg/kg, subcutaneously) 1 h before CAO reduced infarct size in mice [81]. Myocardial infarction promoted a reduction in SOD, catalase, and GPx activity, and increased the malondialdehyde (MDA) level. MI induced ferroptosis [86]. Exendin-4 abolished these alterations [81]. Rats underwent CAO (30 min) and reperfusion (4 h) [41]. Coronary artery ligation resulted in an increase in the myocardial MDA level, ROS generation, and oxidized glutathione (GSSG) content. I/R decreased SOD activity and reduced the glutathione (GSH) level. These findings indicate that MI induced ferroptosis in the myocardium. Pretreatment with exendin-4 (140 ng/kg) abolished these alterations [41]. Consequently, the activation of the GLP1R inhibits ferroptosis. Isolated rat cardiac microvascular endothelial cells (CMECs) were exposed to H/R [107]. H/R upregulated MDA content and downregulated the GSH level, SOD, and GPx activity in CMECs. Liraglutide reversed these alterations [107]. Consequently, GLP1R stimulation inhibits ferroptosis. It is unclear whether the effect is common to the entire class of GLP1 agonists.

Coronary artery ligation (30 min) and reperfusion were carried out in rats [43]. I/R induced an increase in the necroptosis marker levels: phosphorylated receptor-interacting protein kinase 3 (p-RIPK3), and phosphorylated-mixed lineage kinase domain-like protein (p-MLKL) in the infarcted myocardium. Pretreatment with liraglutide reduced infarct size and the p-RIPK3 and p-MLKL levels [43]. These findings demonstrate that GLP1R stimulation abolished the development of necroptosis.

Rats with streptozotocin-induced diabetes were treated with liraglutide for 4 weeks [99]. Liraglutide reduced the expression of a marker of pyroptosis (NOD)-Like Receptor with a Pyrin domain 3 (NLRP3) in the myocardium. The NLRP3 inflammasome, caspase-1, and IL-1β are involved in the development of pyroptosis [87]. The GLP1R agonist inhibited the expression of the NLRP3 inflammasome, cleaved (active) caspase-1, and IL-1β in the myocardium [99]. Consequently, GLP1R stimulation inhibits pyroptosis in the diabetic rat heart.

Autophagy allows cells to survive in adverse conditions, so autophagy has a protective effect in many cases of I/R of the heart [89]. Exendin-4 protected the heart and H9c2 cells against the toxic effect of doxorubicin [108]. This benefit was associated with autophagy stimulation. A study was performed in Zucker diabetic fatty (ZDF) rats (type 2 diabetes) [109]. Chronic administration of liraglutide alleviated diabetic cardiomyopathy. The benefit was accompanied by autophagy stimulation. The autophagy inhibitor chloroquine abolished the cardioprotective effect of liraglutide [109]. Liraglutide upregulated the level of p-AMPK, an autophagy activator, and decreased the level of phosphorylated mammalian target of rapamycin (p-mTOR) kinase, an autophagy inhibitor. An AMPK inhibitor, compound C, partially reversed the cardioprotective effect of liraglutide. Investigators concluded that liraglutide exhibited a cardioprotective effect in ZDF rats through AMPK-dependent autophagy stimulation [109]. It was found that liraglutide stimulated mitophagy in H9_C_2 cells subjected to hypoxia through the sirtuins-11/Parkin/mitophagy pathway [110]. Mice underwent permanent CAO [111]. Chronic administration of the GLP1R agonist DMB prevented the development of adverse cardiac remodeling and stimulated autophagy [111]. Rats underwent transverse aortic constriction (TAC) [112]. Chronic administration of liraglutide reduced cardiac hypertrophy and stimulated autophagy [112]. The human ventricular cardiomyocyte cell line (AC16) was exposed to H/R [113]. H/R induced autophagy. Pretreatment with exendin-4 inhibited autophagy [113].

Thus, the activation of the GLP1R increases cardiomyocyte viability in adverse conditions. In many cases, this effect is associated with stimulation of autophagy and mitophagy [108,110,111,112]. Only He et al. found that GLP1R stimulation can inhibit autophagy [113].

In summary, it was found that the activation of the GLP1R alleviated apoptosis, necroptosis, ferroptosis, and pyroptosis (Figure 1). These effects were accompanied by the cardioprotective effect of GLP1R agonists. The activation of the GLP1R augmented autophagy. This effect was also associated with the cardioprotective effect of GLP1R agonists. Liraglutide now has the largest experimental evidence base among the registered drugs.

## 6. The Infarct-Limiting Effect of Dipeptidyl Peptidase-4 (DPP-4) Inhibitors

An increase in cardiac tolerance to the impact of I/R can be achieved not only by using exogenous agonists but also by using DPP-4 inhibitors [31,34,37]. Dipeptidyl peptidase-4 hydrolyzes GLP-1(7–36) and GLP-1(7–37) to biologically inactive peptides, so its inhibition leads to an increase in the endogenous GLP-1(7–36) and GLP-1(7–37) levels and enhances the effect of exogenous peptide GLP1R agonists [9,23].

Dogs were subjected to CAO (90 min) and reperfusion (6 h) [31]. The DPP-4 inhibitor alogliptin (3 mg/kg/day per os) was administered for 4 days before I/R. The inhibitor reduced infarct size by 60% and suppressed cardiomyocyte apoptosis [31]. As mentioned above, liraglutide reduced infarct size and improved contractility of the heart in reperfusion [34]. The DPP-4 inhibitor linagliptin was administered to rats for 5 days before CAO. Linagliptin had cardioprotective effects similar to those of liraglutide. In addition, linagliptin augmented the cardioprotective effect of the GLP1R agonist liraglutide [34].

The DPP-4 inhibitor MK-0626 (1 mg/kg/day) was administered to mice with permanent CAO for 4 weeks [37]. The drug increased exercise tolerance in mice with MI, which indirectly indicates a decrease in postinfarction cardiac remodeling [37]. Alogliptin (2 or 20 mg/kg/day per os) was administered to rabbits for 7 days before CAO (30 min) and reperfusion (48 h) [40]. Alogliptin dose-dependently reduced infarct size and increased the plasma GLP-1 level [40]. The NO-synthase (NOS) inhibitor completely abolished the infarct-limiting effect of alogliptin. Exendin (9–39) significantly reduced, but did not eliminate, the infarct-reducing effect of alogliptin [40].

These data indicate that pretreatment with DPP-4 inhibitors increases cardiac tolerance to I/R. This effect is dependent on NOS activation and, in part, on an increase in the endogenous GLP-1 level and GLP1R stimulation.

## 7. GLP-1 Prevents the Development of Adverse Remodeling of the Heart

Mice underwent permanent coronary artery ligation [114]. Exendin-4 (0.1 mg/kg intraperitoneally) was injected at 48 h after CAO, then daily for 2 weeks. It was found that exendin-4 increased left ventricular ejection fraction (LVEF). Chronic administration of the GLP1R agonist improved contractility of the heart isolated from mice with MI [114]. Exendin-4 prevents the development of LV hypertrophy, cardiomyocyte hypertrophy, reduced collagen content in the LV, and increased survival of mice with MI. The GLP1R agonist protects H9c2 cardiomyoblasts against H_2_O_2_-induced oxidative stress and prevents H_2_O_2_-induced MPT pore opening [114].

Mice were subjected to permanent CAO [115]. Exendin-4 was administered at a dose of 25 nmol/kg/day by osmotic minipump for 4 weeks. This polypeptide reduced LV end-diastolic volume and LV end-systolic volume, and increased LVEF. The GLP1R agonist decreased cardiac hypertrophy and cardiomyocyte hypertrophy. Exendin-4 inhibited CAO-induced myocardial inflammation. The GLP1R agonist (0.1 µmol/L) alleviated the development of phenylephrine-induced rat ventricular H9c2 cardiomyoblast hypertrophy. Exendin-4 mitigated doxorubicin-induced apoptosis of mouse atrial HL-1 cardiomyocytes [115]. Exendin-4 activated (phosphorylated) Akt and inhibited (phosphorylated) glycogen synthase kinase-3β (GSK-3β) in mice with MI. Polypeptide decreased the expression of fibroblast growth factor-2 (FGF-2) mRNA and matrix metalloprotease-9 (MMP-9) mRNA which participate in extracellular matrix formation and myocardial remodeling. It was surprising that the GLP1R agonist stimulated the expression of proinflammatory cytokines IL-1β and MCP-1 [115]. Consequently, the GLP1R agonist exendin-4 mitigated adverse postinfarction remodeling of the heart by inhibiting FGF-2 and MMP-9 production, stimulating Akt, and inhibiting GSK-3β. It should be noted that Akt kinase activation increases cardiac tolerance to I/R [83]. Inhibition of GSK-3β also augments cardiac resistance to I/R [83].

GLP-1(9–36) was infused to mice with permanent CAO at a rate of 25 nmol/kg/day for four weeks [33]. The indices were assessed 4 weeks after CAO. Peptide did not affect infarct size and cardiac hypertrophy and did not increase LVEF. Consequently, it did not affect postinfarction remodeling. GLP-1(9–36) is a metabolically inactive analog of GLP-1(7–36) [33]. It did not appear to activate the GLP1R and, as a result, did not affect cardiac remodeling. Exendin-4 (10 μg/kg/day, intraperitoneally) was administered to rats with permanent CAO for 4 weeks [39]. The GLP1R antagonist exendin (9–39) (200 μg/kg/day, intraperitoneally) was administered to rats for 4 weeks. Exendin-4 reduced infarct size, prevented cardiac hypertrophy and cardiomyocyte hypertrophy, improved contractility of the myocardium, and reduced interstitial cardiac fibrosis. Exendin (9–39) abolished these beneficial effects of exendin-4 [39]. Consequently, the chronic activation of the GLP1R prevents adverse postinfarction cardiac remodeling.

Rats underwent permanent CAO [116]. Liraglutide (1.2 mg/kg) was administered daily for 4 weeks. Cardiac magnetic resonance imaging was performed after 4 weeks. Liraglutide did not affect adverse remodeling of the heart. Investigators did not explain why liraglutide did not alleviate remodeling.

Liraglutide was administered intravenously at a dose of 0.09 and 0.18 mg/kg for 28 days after permanent CAO in rats [110]. Liraglutide at a dose of 0.18 mg/kg alleviated apoptosis and fibrosis, and downregulated the expression of caspase-3, MMP9, transforming growth factor-β (TGF-β), IL-6, MCP-1, sirtuin-1, and poly(ADP-ribose) polymerases (PARP) [110].

Mice were subjected to permanent coronary artery ligation and treated with the GLP1R agonist DMB (0.4 nmol/kg intraperitoneally) [111]. Administration was repeated three times a week, totaling six injections. Cardiac function was assessed by echocardiography on days 14 and 28. DMB mitigated cardiac fibrosis and improved contractility of the heart. The GLP1R agonist increased the expression of autophagy markers (LC3-II and p62) in the infarcted myocardium. DMB increased mitophagy markers (PINK1, optineurin, and BNIP3) in the mitochondrial fraction. The benefit of DMB was lost in Parkin knockout mice, indicating that Parkin-mediated mitophagy is involved in the cardioprotective effect of DMB [111]. The main defect of the study is the small groups of animals (n = 4), which casts doubt on the significance of these findings.

Rats underwent permanent CAO [117]. A total of 10 rats were used in each group. Exendin-4 was injected (25 nmol/kg/day = 105 µg/kg/day intraperitoneally) for 6 weeks. Exendin-4 increased LV systolic pressure (LVSP), reduced LVEDP, and increased the rate of contraction and relaxation [117]. Exendin-4 upregulated the expression of Bcl-2, an antiapoptotic protein, as well as the silent information regulator-1 (SIRT1), an intracellular survival molecule, and downregulated the expression of the PARP1 DNA repair enzyme. In addition, the GLP1R agonist reduced ROS generation in myocardial tissue. Exendin-4 decreased TNF-α and IL-6 content in the myocardium. Polypeptide downregulated the expression of transcription factor nuclear factor-κB (NF-κB p65) and lamine B1 which are involved in the regulation of gene expression [117]. Exendin-4 reduced collagen I/III and MMP2/9 levels in the infarcted myocardium. Chronic administration of EX527, a SIRT1 inhibitor, at a dose of 5 mg/kg twice per week intraperitoneally abolished the exendin-induced benefit [117]. Investigators concluded that exendin-4 prevents adverse postinfarction cardiac remodeling and mitigates inflammation by activating SIRT1-induced inhibition of PARP1 [117].

Thus, a course of administration of GLP1R agonists could be useful for preventing the formation of adverse postinfarction cardiac remodeling. GLP1R agonists partially abolished the development of adverse remodeling through SIRT1-induced inhibition of PARP1, downregulation of NF-κB, FGF-2, and MMP2/9, activation of Akt, inhibition of GSK-3β, and the stimulation of autophagy and mitophagy.

Can DPP-4 inhibitors prevent adverse postinfarction remodeling? Rats underwent CAO without reperfusion [118]. Cardiac function was measured by echocardiography 12 weeks after CAO. Rats were treated with the DPP-4 inhibitor vildagliptin (15 mg/kg/day) in drinking water. Vildagliptin increased the plasma GLP-1 level three-fold. Vildagliptin did not improve contractility and had no effect on cardiomyocyte size or capillary density after MI. The investigators concluded that vildagliptin increased the plasma GLP-1 level but did not prevent the development of postinfarction cardiac remodeling [118]. The next study was performed in wild-type mice and heterozygous (Eng^+/−^) mice with knockout of the gene encoding endoglin, a transforming growth factor-β (TGF-β) co-receptor [119]. Mice underwent permanent CAO. The DPP-4 inhibitor diprotin A was administered (55 μg/kg/day) for 5 and 14 days. Endoglin knockout promoted an increase in infarct size. Diprotin A had no effect on infarct size in wild-type mice and reduced infarct size in Eng^+/−^ mice. Chronic administration of diprotin A did not improve contractility of the heart both in wild-type mice and Eng^+/−^ mice [119]. However, the DPP-4 inhibitor increased capillary density in infarct border zone, 14 days after CAO. Consequently, inhibition of DPP-4 partially reversed post-infarction myocardial remodeling.

Can GLP1R agonists mitigate adverse overload-induced remodeling? GLP-1 (1.5 pmol/kg/min subcutaneously) was continuously infused to SHHF rats for 3 months by osmotic minipump [91]. GLP-1 increased the survival of rats from 44% in the control group to 72% in the GLP-1 treated group. Chronic infusion of GLP-1 increased LVSP and the relaxation rate of the isolated rat hearts but had no effect on LV developed pressure (LVDP) and the contraction rate. Treatment with GLP-1 increased stroke volume, cardiac output, and LVEF in vivo measured by echocardiography. GLP-1 reduced apoptosis of cardiomyocytes and triggered phosphorylation of Akt [91]. Consequently, the chronic activation of the GLP1R partially abolished hypertension-induced adverse remodeling of the heart apparently through Akt stimulation. Adverse overload-induced remodeling of the heart was induced in rats by partial TAC [120]. Adult male DPP-4-deficient mutant rats were included in the study. TAC induced adverse myocardium remodeling and decreased the circulating GLP-1 level. Exendin-4 was administered at a dose of 10 μg/kg/day intraperitoneally for 14 or 50 days after TAC. Treatment with exendin-4 improved the contractile function of the heart. Chronic administration of the GLP1R antagonist exendin (9–39) reversed the exendin-induced benefit. The GLP1R agonist reduced the expression of NOX1, NOX2, MMP-9, NFκB, and TNF-α, and inhibited oxidative stress. The GLP1R antagonist exendin (9–39) partially reversed but did not abolish this benefit [120]. It is possible that a dose of exendin (9–39) is not enough to completely block all GLP1Rs. Exendin-4 reduced the proapoptotic protein Bax, a phosphorylated suppressor of mothers against decapentaplegic 3 (p-Smad3) protein and TGF-β levels. Exendin (9–39) partially reversed the effect. Smad3 and TGF-β are involved in the development of cardiac fibrosis [120]. Therefore, inhibition of their expression could participate in the cardioprotective effect of exendin-4. Investigators concluded that exendin-4 effectively suppressed TAC-induced adverse cardiac remodeling [120].

Rats underwent TAC [112]. It was used in small groups of animals (n = 6). Liraglutide was injected twice daily at a dose of 0.3 mg/kg subcutaneously for 16 weeks. Liraglutide reduced cardiac hypertrophy, decreased cardiomyocyte hypertrophy, and augmented the plasma GLP-1 level and GLP1R expression in myocardial tissue. Polypeptide inhibited phosphorylation of mTOR and synthesis of collagen I/III in the heart [112]. The GLP1R agonist increased the LC3-II/LC3-I ratio and the Beclin-1 level and downregulated p62 content in myocardial tissue. These alterations are typical manifestations of autophagy stimulation [89]. The activation of (phosphorylation) mTOR kinase promoted inhibition of autophagy [89]. Liraglutide reduced the p-mTOR level in the myocardium. Therefore, it could be proposed that liraglutide alleviated overload-induced adverse remodeling through inhibition of mTOR. It should be noted that in many cases autophagy stimulation increases cardiac tolerance to I/R [89]. The data obtained by Zheng et al. [112] demonstrated that autophagy activation can prevent the development of adverse remodeling of the heart.

Rats were subjected to TAC for 16 weeks [121]. Liraglutide was administered at a dose of 0.3 mg/kg/twice a day, and the ATP-sensitive K^+^ channel (K_ATP_ channel) blocker glibenclamide was used at a dose of 5 mg/kg/day. Liraglutide reduced cardiac hypertrophy, cardiomyocyte hypertrophy, and a number of apoptotic cells in the myocardium. Polypeptide improved acetylcholine-triggered aortic relaxation. Liraglutide upregulated the expression of Kir6.2/SUR2 subunits of the K_ATP_ channel in the myocardium, cardiac microvasculature, aortic endothelium, and vascular smooth muscle. Polypeptide increased Bcl2 expression and the ATP level in the heart, and reduced the serum troponin I concentration. The GLP1R agonist improved the contractility of the heart. Glibenclamide completely abolished the antiapoptotic effect of liraglutide, partially abolished its effect on cardiomyocyte hypertrophy, as well as its cytoprotective effect, and the inotropic effect of the polypeptide [121]. Consequently, liraglutide prevents the development of TAC-induced adverse remodeling of the heart partially through K_ATP_ channel opening. However, GLP-1(7–36) inhibits K_ATP_ channels in β-cells of the islets of Langerhans, which leads to increased insulin secretion [122]. It should be noted that K_ATP_ channels in cardiomyocytes and in β-cells are different [123]. Apparently, this feature of K_ATP_ channels allows GLP-1 to block them in β-cells and activate them in cardiomyocytes.

Angiotensin II participates in the pathogenesis of adverse remodeling of the heart [124]. The isolated murine cardiac fibroblasts were incubated with different concentrations of glucose or angiotensin II for 24 h [34]. High concentrations of glucose (≥40 mmol/L) and angiotensin II (≥10^−6^ mol/L) induced the expression of fibrotic molecules (fibronectin and collagen-1, -3, and -4). Liraglutide inhibited collagen synthesis in cardiac fibroblasts via the activation of the ERK/NF-κB/pathway [34]. Consequently, the activation of the GLP1R in cardiac fibroblasts could be mediated by GLP1R agonists-triggered inhibition of adverse overload-induced remodeling. Hypertension and adverse remodeling were induced by subcutaneous infusion of angiotensin II (100 ng/kg/min) for 4 weeks in mice by osmotic minipumps [125]. Liraglutide was injected at a dose of 400 µg/kg/day intraperitoneally for 4 weeks. Liraglutide decreased blood pressure (BP) and reduced angiotensin-1 receptor (AT1R) expression, as well as ROS production in the myocardium [125]. In a study with isolated cardiac fibroblasts, it was found that liraglutide inhibited angiotensin-induced collagen synthesis, the upregulation of AT1R, and ROS production [125]. It could be hypothesized that GLP1R stimulation in cardiac fibroblasts can alleviate overload-induced adverse remodeling.

Adverse cardiac remodeling can develop in diabetes [126]. Streptozotocin-induced diabetes was induced in mice [127]. Exendin-4 and insulin were administered for 4–12 weeks. Administration of both exendin-4 and insulin improved metabolic indices in mice with diabetes. However, only exendin-4 improved cardiac contractility and alleviated interstitial cardiac fibrosis [127]. These findings demonstrate that chronic stimulation of the GLP1R can mitigate diabetes-induced adverse remodeling of the heart. Rats with streptozotocin-induced diabetes were treated with the DPP-4 inhibitor sitagliptin (10 mg/kg/day for 6 weeks) [128]. Sitagliptin reduced cardiac hypertrophy, serum glucose, GLP-1, CK-MB, and cardiac troponin-I levels. The DPP-4 inhibitor decreased collagen I and collagen III mRNA content in myocardial tissue. These effects were associated with an increase in the antiapoptotic protein Bcl2 level and a decrease in proapoptotic protein Bax content in the heart. Sitagliptin increased p-Akt and p-AMPK expression and also decreased the p-p38 kinase level in the myocardium of diabetic rats [128]. These findings demonstrated that DPP-4 inhibition reversed streptozotocin-induced adverse remodeling of the heart.

In summary, these data convincingly demonstrate that the chronic activation of the GLP1R prevents the development of adverse remodeling of the heart in myocardial infarction, hypertension, and diabetes. It should be noted that most data obtained now are for liraglutide. DPP-4 inhibitors could prevent adverse diabetes-induced myocardial remodeling. The cardioprotective effect of GLP1R agonists is associated with the activation of Akt, SIRT1, inhibition of GSK-3β, downregulation of ROS generation, NOX1, NOX2, NF-κB, TNF-α, IL-6, MMP-2/9, FGF-2, p-mTOR, AT1R, and Smad3 expression. Autophagy is involved in the cardioprotective effect of GLP1R agonists in animals with permanent CAO and partial transverse aortic constriction. Chronic stimulation of the GLP1R can alleviate diabetes-induced adverse remodeling of the heart.

## 8. GLP-1 Protects against Diabetic Cardiomyopathy

As mentioned above, chronic administration of liraglutide inhibited apoptosis of cardiomyocytes in rats with streptozotocin-induced diabetes (type 1 diabetes) [96]. Chronic treatment with exendin-4 prevented the formation of streptozotocin-induced diabetic cardiomyopathy in mice [127]. The chronic application of liraglutide alleviated diabetic cardiomyopathy in ZDF rats with type 2 diabetes [109]. Administration of the DPP-4 inhibitor sitagliptin (10 mg/kg/day) for 6 weeks mitigated the development of streptozotocin-induced diabetic cardiomyopathy in rats [128]. The DPP-4 inhibitor alogliptin (12.5 mg/kg/day for 12 weeks) delayed the formation of streptozotocin-induced contractile dysfunction in rabbits [129]. The GLP-1 analog exendin-4 and the DPP-4 saxagliptin improved contractile function and prevented cardiac remodeling in streptozotocin-induced diabetic mice [130]. Type 2 diabetes was modeled in mice by a high-fat diet for 4 weeks [131]. Type 1 diabetes was modeled in mice by injection of streptozotocin [131]. Chronic application of exendin-4 (24 nmol/kg/day) improved contractility of the heart in mice with type 2 diabetes but not in mice with type 1 diabetes [131].

Old ZDF rats with type 2 diabetes on a high-salt diet were treated with vehicle or liraglutide (0.1 mg/kg/day, subcutaneously) for 8 weeks [132]. Liraglutide reduced systolic blood pressure and improved acetylcholine-induced vasodilation in the small arteries. In addition, polypeptide downregulated NF-κB, IL-1β, TGF-β1, and osteopontin expression in the left ventricle [132]. Streptozotocin-induced diabetic rats were treated with exendin-4 for 3 months [98]. Exendin-4 improved contractile function, reduced BNP mRNA and collagen content in the myocardium, and inhibited apoptosis of cardiomyocytes [98]. Consequently, exendin-4 alleviated diabetic cardiomyopathy. Rats with streptozotocin-induced diabetes were treated with liraglutide for 4 weeks [99]. Liraglutide improved contractility of the heart, inhibited apoptosis, alleviated cardiac fibrosis, and increased the myocardial p-AMPK, p-Akt, p-ERK, phosphorylated signal transducer and activator of transcription 3 (p-STAT3) levels [99]. The peptide GLP1R and glucagon receptor agonist ZP2495 improved contractility of the heart and inhibited cardiomyocyte apoptosis in diabetic mice db/db (type 2 diabetes) with CAO and reperfusion [102].

These findings convincingly demonstrated that chronic administration of GLP1R agonists alleviated the development of diabetic cardiomyopathy in animals with type 1 and type 2 diabetes.

## 9. GLP-1 Protects against Oxidative Stress

Oxidative stress is involved in reperfusion cardiac injury [133]. Stimulation of the GLP1R prevents the development of reperfusion damage of the heart. Therefore, it could be hypothesized that GLP1R agonists can protect the heart against oxidative stress.

Pigs underwent ventricular fibrillation followed by resuscitation [134]. GLP-1 was infused for 4 h after the onset of resuscitation. GLP-1 reduced the plasma concentration of 8-epi-prostaglandin F_2α_ isoprostane, a marker of oxidative stress [134]. H9c2 cells were exposed to oxidative stress induced by H_2_O_2_ [81]. Pretreatment with exendin-4 (1 nmol/L) increased cell viability and inhibited H_2_O_2_-triggered ROS generation [81]. Pretreatment with exendin-4 (10 μg/kg/day for 10 days) prevents the development of oxidative stress in the infarcted rat myocardium [81]. In addition, pretreatment with exendin-4 alleviated oxidative stress induced by H/R in H9c2 cells [81]. Rat cardiomyocytes with incubated with 21 mM glucose [135]. The high-glucose level induced ROS production. GLP-1 inhibited ROS generation [135]. As mentioned above, oxidative stress in isolated neonatal rat cardiomyocytes was induced by H_2_O_2_ [95]. Exendin-4 inhibited apoptosis and ROS production through the activation of Epac [95].

As already mentioned above, apoptosis of isolated neonatal rat cardiomyocytes was induced by TNF-α [100]. Exendin-4 inhibited ROS production and apoptosis of rat cardiomyocytes [100]. Liraglutide inhibited apoptosis of cardiomyocytes in the myocardium of rats with streptozotocin-induced diabetes [96]. This antiapoptotic effect of polypeptide was accompanied by a decrease in ROS production in myocardial tissue. Incubation of CMECs with H_2_O_2_ induced apoptosis of these cells [107]. Liraglutide inhibited apoptosis of CMECs. H/R of CMECs triggered ROS production. Liraglutide inhibited H/R-induced oxidative stress. Xanthine oxidase (XO) siRNA exhibited the same effect in H/R. XO synthesizes superoxide radicals in rat cardiomyocytes [136]. The investigators proposed that liraglutide inhibits XO, which leads to a reduction in ROS production. Evidence was obtained that liraglutide downregulates ROS generation through the activation of the GLP1R/PI3K/Akt/survivin pathway [107]. Liraglutide was administered to mice at a dose of 10 μg/kg subcutaneously for 5 days before CAO and reperfusion [34]. Liraglutide reduced infarct size and downregulated ROS production and NADPH oxidase-2 (NOX2) expression in the infarcted myocardium [34]. NOX2 synthesized superoxide radicals in cardiomyocytes [136]. It was suggested that the downregulation of NOX2 is involved in liraglutide-induced inhibition of ROS generation in I/R of the heart. Liraglutide mitigated H/R-induced apoptosis of H9c2 cells and cardiomyocytes and inhibited ROS production [101].

Oxidative stress in isolated rat cardiomyocytes was induced by incubation with H_2_O_2_ [137]. It was found that liraglutide prevents the development of H_2_O_2_-induced Ca^2+^ overload cardiomyocytes [137]. A course of administration of liraglutide to mice with permanent CAO augmented the expression of the transcription factor nuclear factor erythroid 2-related factor 2 (Nrf2) [44]. Nrf2 induced transcription of genes encoding proteins of antioxidant protection [138]. The GLP1R agonist ZP2495 blocked MPT pore opening and mitochondria ROS production in diabetic mice db/db subjected to CAO and reperfusion [102]. Exendin-4 inhibited ROS production in the myocardium of rats with streptozotocin-induced diabetes [98]. Exendin-4 reduced the MDA and GSSG levels, as well as ROS generation, and increased GSH in the infarcted myocardium of rats [41,117]. Liraglutide inhibited angiotensin-induced ROS production in mice [125].

GLP1R agonists inhibit ROS generation in the myocardium of animals and augment the tolerance of isolated cardiomyocytes to oxidative stress. Evidence was obtained that GLP1R stimulation inhibits free radical generation through the downregulation of NOX2 expression and the upregulation of Nrf2. GLP1R agonists inhibit ROS production through an increase in Epac expression and the activation of the GLP1R/PI3K/Akt/survivin pathway (Figure 2). It was found that the activation of the GLP1R increased the tolerance of isolated cardiomyocytes to oxidative stress through the inhibition of Ca^2+^ overload.

## 10. GLP-1 Decreases the Proinflammatory Cytokine Levels in the Myocardium 

Exendin-4 downregulated bacterial lipopolysaccharide-induced TNF-α mRNA in H9c2 cells [139]. As mentioned above, liraglutide reduced the serum IL-6 and TNF-α in mice with CAO and reperfusion [34]. Polypeptide decreased IL-6 and MCP-1 expression in the infarcted myocardium [34]. Chronic application of exendin-4 downregulated TNF-α expression in the myocardium of rats with TAC [120]. Rats underwent permanent CAO [117]. Chronic administration of exendin-4 decreased TNF-α and IL-6 content in myocardial tissue [117]. Rats with streptozotocin-induced diabetes were treated with liraglutide for 4 weeks [99]. Chronic GLP1R activation inhibited IL-1β expression in the myocardium of rats with streptozotocin-induced diabetes [99]. Consequently, chronic GLPR stimulation reduced IL-1β, TNF-α, IL-6, and MCP-1 expression in the myocardium.

## 11. The Role of GLP-1 in the Cardioprotective Effect of Remote Conditioning

Remote postconditioning is an adaptive phenomenon that ensures cardiac tolerance to long-term I/R due to short-term I/R of a remote organ in cardiac reperfusion [140]. Remote preconditioning (RPre) is an adaptive phenomenon that ensures the heart’s resistance to long-term I/R due to short-term I/R of a remote organ before cardiac ischemia [141]. Remote perconditioning (RPer) is an increase in the heart’s resistance to long-term I/R due to short-term I/R of a remote organ during CAO [35].

Rats were subjected to CAO (30 min) and reperfusion (120 min) [35]. RPer was modeled using occlusion (15 min) and reperfusion of the femoral artery 10 min after the onset of I/R. RPre was performed by occlusion (15 min) and reperfusion of the femoral artery 25 min before the onset of I/R. Both interventions contributed to a 50% reduction in infarct size. The GLP1R antagonist exendin (9–39) (50 μg/kg intravenously) completely abolished the infarct-limiting effect of RPre and PPer. Bilateral cervical vagotomy also eliminated the adaptive increase in cardiac tolerance to I/R. The GLP1R agonist exendin-4 (5 μg/kg intravenously) mimics the cardioprotective effect of remote conditioning, and reduced infarct size by 50% [35]. These data demonstrate that GLP1R and n. vagus are involved in the cardioprotective effect of RPre and RPer.

## 12. The Signaling Mechanism of the Cardioprotective Effect of GLP1R Agonists

It has been found that the following protein kinases are involved in the cardioprotective effect of ischemic pre- and postconditioning: phosphatidylinositol 3-kinase (PI3K), mitogen-activated protein kinase kinase-1/2 (MEK1/2), ERK1/2. AMPK, PKC, PKG, and Akt kinase [83]. The activation of these kinases increases cardiac tolerance to I/R [83]. Stimulation of NOS and heme oxygenase (HO-1) also increases cardiac resistance to I/R [83]. Both enzymes participate in conditioning [83]. cGMP and sirtuins are regulatory proteins involved in the cardioprotective effect of conditioning [83]. In contrast, the activation of GSK-3β aggravates I/R injury of the heart [83]. K_ATP_ channels, big conductance Ca^2+^-activated K^+^ channels (BK_Ca_ channels), and MPT pores are involved and are hypothetical end-effectors of ischemic pre- and postconditioning [18,83,123]. It has been found that K_ATP_ channel opening, BK_Ca_ channel opening, and MPT pore closing promote an increase in cardiac tolerance to I/R [123,142,143].

GLP-1 at a final concentration of 3 nmol/L protected the perfused isolated rat heart against local I/R [24]. Exendin (9–39), the PI3K inhibitor LY294002, the MEK1/2 and ERK1/2 inhibitor U0126, and the PKA inhibitor Rp-cAMP abolished the infarct-reducing effect of GLP-1 [24]. Blockade of mTOR kinase also eliminated the cardioprotective effect of GLP1 in vitro [25]. PI3K, MEK1/2, and ERK1/2 are involved in the antiapoptotic effect of GLP-1 in a study with isolated rat cardiomyocytes subjected to H/R [92]. Liraglutide upregulated cAMP content in myocardial tissue. Consequently, PI3K, MEK1/2, ERK1/2, and PKA are involved in the cardioprotective effect of GLP-1.

Noyan-Ashraf et al. found that a course of administration of the GLP1R agonist liraglutide to mice causes phosphorylation (activation) of Akt and HO-1 in mice with permanent CAO [44]. At the same time, there is an inactivation of GSK-3β and increased Nrf2 expression [44]. Liraglutide upregulated the expression of peroxisome proliferator-activated receptor (PPAR)-β/δ [44]. The activation of this receptor promoted an increase in cardiac tolerance to I/R [144]. In studies performed in isolated cardiomyocytes subjected to H/R, GLP-1(9–36)amide and exendin-4 increase cell survival [46]. Both peptides increased the cAMP level in cardiomyocytes and induced phosphorylation (activation) of Akt, ERK1/2, and transcription factor cAMP response element-binding protein (CREB) [46]. Akt and ERK1/2 inhibitors reversed the cytoprotective effect of GLP1R agonists [46]. LY294002, a PI3K inhibitor, abolished the infarct-limiting effect of exendin-4 [82].

There is evidence that exendin inhibits ferroptosis in the inflamed myocardium [82]. The phosphodiesterase-3 inhibitor cilostazol, which inhibits cAMP hydrolysis and increases the cAMP level in tissues, significantly enhanced the infarct-limiting effect of exendin [28]. The PKA inhibitor H-89 completely abolished the infarct-sparing effect of exendin. Exendin increased the cAMP level and PKA activity in the infarcted myocardium [28]. These data suggest that cAMP and PKA are involved in the infarct-limiting effect of exendin. The combined use of exendin and cilostazol led to the phosphorylation of CREB, Akt, and ERK1/2 [28], which could also be directly related to the cardioprotective effect of exendin. Exendin-4 did not affect infarct size in mice with a knockout of the gene encoding Akt and with a knockout of the gene encoding mitogen-activated protein kinase kinase-3 (MKK3) [32]. Thus, Akt and MKK3 kinases are involved in the infarct-reducing effect of GLP1R agonists. The DPP-4 inhibitor linagliptin (3 mg/kg daily for 5 days) reduced infarct size in mice with a 60 min CAO and reperfusion (3 h) by about 50% [34]. Linagliptin decreased the ROS level and the expression NOX4 which generates superoxide radicals in the infarcted myocardium.

Chronic stimulation of the GLP1R using administration of exendin-4 for 4 weeks prevents adverse postinfarction cardiac remodeling in rats with permanent CAO [39]. Exendin-4 increased GLP1R and GLP-1 expression in the infarcted myocardium. This peptide increased the cGMP level in the myocardium by 2.5-fold, augmented PKG and endothelial NOS (eNOS) expression, and also stimulated phosphorylation of eNOS [39]. In addition, data were obtained that exendin-4 improves Ca^2+^ transport at the level of the sarcoplasmic reticulum of cardiomyocytes, thereby preventing calcium overload of cells. The GLP1R antagonist exendin (9–39) abolished these beneficial effects of exendin-4. It was suggested that the beneficial effect of exendin-4 in post-infarction cardiac remodeling is associated with the activation of the GLP1R/eNOS/cGMP/PKG signaling pathway [39].

Exendin-4 (140 ng/kg) increased the expression of the regulatory protein sirtuin-1 and stimulated phosphorylation (activation) of AMPK [41]. The sirtuin expression inhibitor EX527 (75 mg/kg) abolished the infarct-limiting effect of exendin-4 and suppressed exendin-induced sirtuin-1 expression and phosphorylation of AMPK [41]. Therefore, exendin-4 could have an infarct-limiting effect by upregulating sirtuin-1 expression and the activation of AMPK. Exendin-4 (140 ng/kg) increased antiapoptotic protein Bcl-2 expression and decreased proapoptotic protein Bax expression, decreased the activated caspase-9 level, and activated caspase-3 in the inflamed myocardium [145]. It is possible that exendin-4 could have an antiapoptotic effect, which also contributes to increased cardiac resistance to I/R. The GLP1R agonist liraglutide increased GLP1R expression and caused the activation of Akt and PI3K in the rat myocardium [43]. It was found that the infarct-reducing and antiapoptotic effects of GLP1R stimulation are mediated through the activation of the PKG/PKCε/ERK1/2 pathway [106].

These data indicate that the following kinases are involved in the cardioprotective effect of GLP1R agonists: PKCε, PKA, Akt, AMPK, PI3K, ERK1/2, mTOR, GSK-3β (inactivated), PKG, MEK1/2, and MKK3. In addition to kinases, the following enzymes are involved in the infarct-limiting effect of GLP1R agonists: HO-1 and eNOS (Figure 3). There is evidence that the transcription factors CREB and Nrf2 play an important role in cardioprotection. The cardioprotective effect of GLP1R stimulation is associated with an increase in PPAR-β/δ.

## 13. The Limitations and Side Effects

GLP1R agonists can induce a decrease in body weight [146]. This effect is positive in patients with overweight or obesity. However, this effect will be negative in patients with underweight and cachexia because GLP1R agonists may worsen the state of their health. GLP1R agonists can exhibit side effects in 20–29% of patients [147,148]. These are gastrointestinal side effects including nausea, vomiting, and sometimes diarrhea or abdominal pain [147,148].

## 14. Unresolved Issues

It has been found that remote postconditioning, adaptation to hypoxia, and cold increased cardiac tolerance to I/R [86,141,149]. The question of whether GLP-1 and the GLP1R are involved in the cardioprotective effect of remote postconditioning and adaptation to hypoxia and cold remains to be studied. The β_1_-adrenergic receptor (β_1_-AR) is involved in stress-induced cardiac injury [150]. GLP1R agonists protect the heart against the cardiotoxic effect of the β-AR agonist isoproterenol [151,152]. However, it is unknown whether GLP1R agonists protect the heart against stress-induced cardiac injury. It is unclear whether BK_Ca_ channels are involved in the cardioprotective effect of GLP1R stimulation.

## 15. Conclusions

These findings convincingly demonstrate that the activation of the GLP1R promotes cardiac tolerance to I/R. GLP1R agonists are cardioprotective when used acutely and chronically before ischemia and when used in reperfusion. The latter fact suggests the feasibility of clinical trials of GLP1R agonists in patients with AMI and percutaneous coronary intervention when cardiac reperfusion injury occurs. GLP1R agonists not only reduce infarct size but also restore cardiac contractility in reperfusion. GLP1R agonists prevent the formation of adverse postinfarction cardiac remodeling, which also suggests the feasibility of clinical trials of these drugs in patients who have suffered an AMI. The use of DPP-4 inhibitors is an alternative approach to improve cardiac resistance to I/R. GLP-1 and the GLP1R are involved in the infarct-limiting effect of remote conditioning. It has been shown that the cardioprotective effect of GLP1R agonists involves sirtuin-1, kinases (PKCε, AMPK, PI3K, ERK1/2, PKA, mTOR, GSK-3β, PKG, and MKK3), and a number of other enzymes (HO-1 and eNOS). The transcription factors STAT3, Nrf2, FoxO3, and CREB are involved in the cardioprotective effect of GLP1R agonists.

The cardioprotective effect of GLP1R agonists is associated with NFκB expression. GLP1R stimulation leads to inhibition of apoptosis, ferroptosis, necroptosis, and pyroptosis, which could also be directly related to cardiac resistance to I/R. The activation of the GLP1R augments autophagy which also promotes cardiac tolerance to I/R. Chronic administration of GLP1R agonists alleviated the development of diabetic cardiomyopathy in animals. GLP1R stimulation inhibits ROS generation in the myocardium and increases the tolerance of isolated cardiomyocytes to oxidative stress. GLP1R stimulation inhibits ROS generation through a decrease in NOX2 expression and an increase in Nrf2 expression. GLP1R agonists downregulated ROS generation through the activation of the GLP1R/PI3K/Akt/surviving pathway. GLP1R agonists increased cardiomyocyte tolerance to oxidative stress through inhibition of Ca^2+^ overload. Chronic GLPR stimulation reduced IL-1β, TNF-α, IL-6, and MCP-1 expression in the myocardium, indicating effects on subclinical inflammation. GLPR agonist can trigger K_ATP_ channel opening and MPT pore closing. These effects are involved in the cardioprotective effect of GLPR agonists.

## Figures and Tables

**Figure 1 ijms-25-04900-f001:**
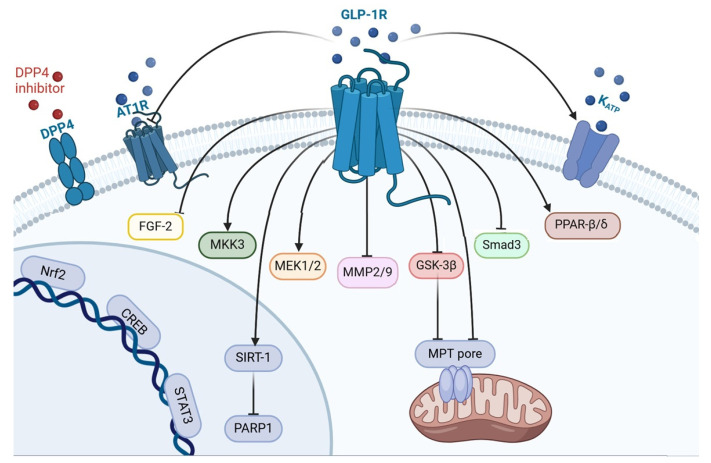
The mechanism of GLP-1-induced prevention of myocardial remodeling. DPP4, dipeptidyl peptidase 4; AT1R, angiotensin-1 receptor; GLP1R, glucagon-like peptide-1 receptor; K_ATP_, ATP sensitive potassium channel; FGF-2, fibroblast growth factor-2; MKK3, mitogen-activated protein kinase kinase-3; MEK1/2, mitogen-activated protein kinase kinase-1/2; MMP2/9, matrix metalloproteinase 2/9; GSK3β, kinase glycogen synthase 3 β; Smad3, Suppressor of Mothers Against Decapentaplegic 3; PPAR-β/δ, peroxisome proliferator-activated receptor; Nrf2, nuclear factor erythroid 2-related factor 2; CREB, cAMP response element-binding protein; STAT3, signal transducer and activator of transcription 3; SIRT-1, silent information regulator-1; PARP1, poly(ADP-ribose) polymerase 1; MPT-pore, mitochondrial permeability transition pore.

**Figure 2 ijms-25-04900-f002:**
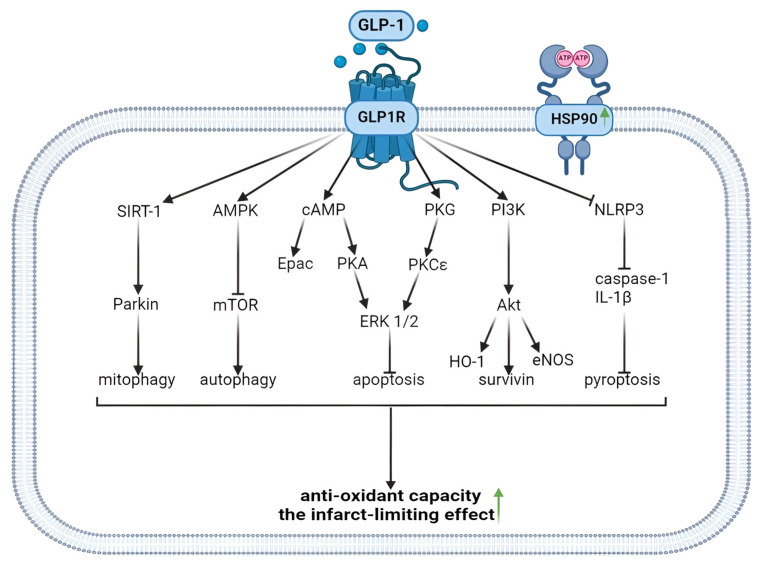
Antioxidant and infarct-limiting effect of GLP-1 receptor activation. Glucagon-like peptide-1 and regulated forms of cell death. GLP1, glucagon-like peptide-1; GLP1R, glucagon-like peptide-1 receptor; HSP90, heat shock protein 90; SIRT-1, silent information regulator-1; AMPK, adenosine monophosphate-activated protein kinase; cAMP, cyclic adenosine monophosphate; PKG, protein kinase G; PI3K, phosphoinositide 3-kinase; NLRP3, (NOD)-Like Receptor with a Pyrin domain 3; mTOR, mammalian target of rapamycin; Epac, exchange protein activated by cAMP-1; PKA, protein kinase A; PKCε, protein kinase Cε; ERK1/2, extracellular signal-regulated kinase; Akt, Akt kinase; IL-1β, interleukin-1 β; HO-1, hemeoxigenase 1; eNOS, endothelial NO-synthase. The upward arrow indicates an increase in antioxidant capacity and the formation of infarct-limiting effect.

**Figure 3 ijms-25-04900-f003:**
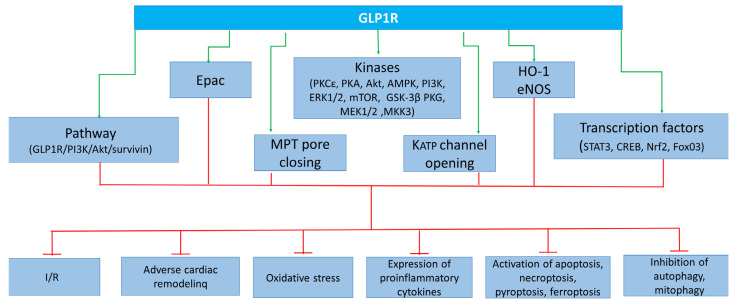
Cardioprotective effect of GLP-1 receptor agonists due to activation of kinases, enzymes, transcription factors, KATP channel opening channels, and MPT pore closing. GLP1R, glucagon-like peptide-1 receptor; Epac, exchange protein activated by cAMP-1; PKCε, protein kinase Cε; PKA, protein kinase A; Akt, Akt kinase; AMPK, adenosine monophosphate-activated protein kinase; PI3K, phosphoinositide 3-kinase; ERK1/2, extracellular signal-regulated kinase; mTOR, mammalian target of rapamycin; GSK3β, kinase glycogen synthase 3 β; PKG, protein kinase G; MEK1/2, mitogen-activated protein kinase kinase-1/2; MKK3, mitogen-activated protein kinase kinase-3; MPT pore, mitochondrial permeability transition pore; K_ATP_, ATP sensitive potassium channel; STAT3, signal transducer and activator of transcription 3; CREB, cAMP response element-binding protein; Nrf2, nuclear factor erythroid 2-related factor 2; FoxO3, Forkhead box O3; I/R, ischemia/reperfusion.

**Table 1 ijms-25-04900-t001:** **Circulating glucagon-like peptides-1.**

GLP-1(7–36)	His-Ala-Glu-Gly-Thr-Phe-Thr-Ser-Asp-Val-Ser-Ser-Tyr-Leu-Glu-Gly-Gln- Ala-Ala-Lys-Glu-Phe-Ile-Ala-Trp-Leu-Val-Lys-Gly-Arg.
GLP-1(7–37)	His-Ala-Glu-Gly-Thr-Phe-Thr-Ser-Asp-Val-Ser-Ser-Tyr-Leu-Glu-Gly-Gln- Ala-Ala-Lys-Glu-Phe-Ile-Ala-Trp-Leu-Val-Lys-Gly-Arg-Gly.

**Table 2 ijms-25-04900-t002:** **Exendin’s structure.**

Exendin-3	His-Ser-Asp-Gly-Thr-Phe-Thr-Ser-Asp-Leu-Ser-Lys-Gln-Met-Glu-Glu- Glu-Ala-Val-Arg-Leu-Phe-Ile-Glu-Trp-Leu-Lys-Asn-Gly-Gly-Pro-Ser- Ser-Gly-Ala-Pro-Pro-Pro-Ser-NH_2_
Exendin-4	His-Gly-Glu-Gly-Thr-Phe-Thr-Ser-Asp-Leu-Ser-Lys-Gln-Met-Glu-Glu- Glu-Ala-Val-Arg-Leu-Phe-Ile-Glu-Trp-Leu-Lys-Asn-Gly-Gly-Pro-Ser- Ser-Gly-Ala-Pro-Pro-Pro-Ser-NH_2_.
Exendin-4 (9–39)	Asp-Leu-Ser-Lys-Gln-Met-Glu-Glu-Glu-Ala-Val-Arg-Leu-Phe-Ile-Glu-Trp- Leu-Lys-Asn-Gly-Gly-Pro-Ser-Ser-Gly-Ala-Pro-Pro-Pro-Ser-NH_2_

## Data Availability

The datasets analyzed during the current study are available in the PubMed repository.

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
