# Peer review of "Peptides Are Cardioprotective Drugs of the Future: The Receptor and Signaling Mechanisms of the Cardioprotective Effect of Glucagon-like Peptide-1 Receptor Agonists"

_ijms, 2024, doi:10.3390/ijms25094900_

Round 1

Reviewer 1 Report

Comments and Suggestions for Authors

This review recounts a large number of studies in H9c2 cells and animals that have investigated the role of GLP1 receptor (GLP1R) stimulation in situations that put stress on the cells such as ischemia, reperfusion or increased glucose levels. It appears that GLP1R stimulation acts protective in the vast majority of these studies. DPP4 inhibitors were not always able to reproduce the effects of GLP1R stimulation. A major weakness of many of these studies is that the GLP1R stimulation was performed before the stress signal. For example, animals were treated with GLP1R agonists before exposing them to cardiac ischemia. This is a rather unlikely setting for a heart attack (unless the patient is diabetic and has received GLP1R agonists before the heart attack).

While the title claims that “peptides are cardioprotective drugs of the future” the authors refer to at least one study with a small molecule allosteric agonist at the GLP1R. It is likely that either orthosteric or allosteric non-peptidic compounds will be developed for the GLP1R in the not-too-distant future. I would therefore prefer if the first sentence was omitted from the title.

With respect to the G-protein coupling of GLP1R that is described on page 2, it has recently been shown that the GLP1R couples to all four G protein families, i.e. Gs, Gi/o, Gq/11 and G12/13 (Avet C, Mancini A, Breton B, Le Gouill C, Hauser AS, Normand C, et al. (2022). Effector membrane translocation biosensors reveal G protein and βarrestin coupling profiles of 100 therapeutically relevant GPCRs. Elife 11: e74101.). It should be emphasized that Gs is the preferred partner of the GLP1R while the GLP1R is least likely to couple to G12/13. I do not think that it is relevant that Gi/o and Gq/11 are involved in the cardioprotective effects of some G-protein-coupled receptors since they can also mediate cardio-damaging effects.

Several experiments appear to use relatively high GLP1R agonist doses. Could the authors elaborate on how would these doses compare to doses used in humans (if the drug is used in humans; obviously not possible for investigational compounds)?

Maybe the authors could also include a table about the various compounds or describe some of them in more detail. For example, I did not know either ZP2495 (a peptidic agonist at the GLP1 and glucagon receptor) or DMB (a small molecule allosteric GLP1R agonist).

Comments on the Quality of English Language

The manuscript would benefit from proofreading by a native speaker. I suggest the following improvements (this list is not exhaustive):

line 42: instead of “polypeptide increases …” -> “GLP1 increases …”

line 49: instead of “The GLP1R is the G protein-coupled receptor” -> “The GLP1R is a G protein-coupled receptor”

line 80: instead of “In 1992, polypeptide exendin-4 was isolated …” -> “In 1992, the polypeptide exendin-4 was isolated …”

line 86: instead of “In 1991, it was found the selective peptide GLP1R antagonist.” -> “In 1991, a selective peptide GLP1R antagonist was found.”

line 105: instead of “… for example, resistant to enzyme hydrolysis erythropoietin …” -> “… for example erythropoietin …”

line 135-136: instead of “it was reported on the neuroprotective effects of liraglutide” -> “neuroprotective effects of liraglutide were reported”

line 231: instead of “exendin-4 increased a number of surviving cardiomyocytes” -> “exendin-4 increased the number of surviving cardiomyocytes”

line 312: the sentence “H9c2 cells exposed to H/R” seems to be incomplete

line 337: instead of “However, investigators used small groups” -> “However, the investigators used small groups”

line 516: instead of “Compund mitigated …” -> “DMB mitigated …”

line 523-524: instead of “It was used 10 rats in each group. Exendin (25 nmol/kg/day = 105 µg/kg/day intraperitoneally) for 6 weeks.” -> “10 rats in each group were injected intraperitoneally (25 nmol/kg/day = 105 µg/kg/day) with exendin-4 for 6 weeks”

line 631: instead of “Adverse cardiac remodeling can be developed in diabetes.” -> “Adverse cardiac remodeling can develop in diabetes.”

line 634: instead of “However, exendin-4 only improved …” -> “However, only exendin-4 improved …”

line 777: Instead of “It has been found that following protein kinases are …” -> “It has been found that the following protein kinases are …”

line 824: what is “course administration”?

Author Response

This review recounts a large number of studies in H9c2 cells and animals that have investigated the role of GLP1 receptor (GLP1R) stimulation in situations that put stress on the cells such as ischemia, reperfusion or increased glucose levels. It appears that GLP1R stimulation acts protective in the vast majority of these studies. DPP4 inhibitors were not always able to reproduce the effects of GLP1R stimulation. A major weakness of many of these studies is that the GLP1R stimulation was performed before the stress signal. For example, animals were treated with GLP1R agonists before exposing them to cardiac ischemia. This is a rather unlikely setting for a heart attack (unless the patient is diabetic and has received GLP1R agonists before the heart attack).

While the title claims that “peptides are cardioprotective drugs of the future” the authors refer to at least one study with a small molecule allosteric agonist at the GLP1R. It is likely that either orthosteric or allosteric non-peptidic compounds will be developed for the GLP1R in the not-too-distant future. I would therefore prefer if the first sentence was omitted from the title.

New title was suggested by other reviewers.

With respect to the G-protein coupling of GLP1R that is described on page 2, it has recently been shown that the GLP1R couples to all four G protein families, i.e. Gs, Gi/o, Gq/11 and G12/13 (Avet C, Mancini A, Breton B, Le Gouill C, Hauser AS, Normand C, et al. (2022). Effector membrane translocation biosensors reveal G protein and βarrestin coupling profiles of 100 therapeutically relevant GPCRs. Elife 11: e74101.). It should be emphasized that Gs is the preferred partner of the GLP1R while the GLP1R is least likely to couple to G12/13. I do not think that it is relevant that Gi/o and Gq/11 are involved in the cardioprotective effects of some G-protein-coupled receptors since they can also mediate cardio-damaging effects.

We cited your article and PMID: 38461904. It has been proven that Gi/o and Gq/11 is involved in the cardioprotective effect of adenosine, bradykinin, opioids, cannabinoids and angiotensin-(1-7). See review articles, please PMID: 14506302 PMID: 27197922 PMID: 26487546 PMID: 38311344 PMID: 26140711 

Several experiments appear to use relatively high GLP1R agonist doses. Could the authors elaborate on how would these doses compare to doses used in humans (if the drug is used in humans; obviously not possible for investigational compounds)?

 We discussed all preclinical studies of GLP1R agonists inducing cardioprotective effect. Clinical trials of GLP1R agonists as cardioprotective drugs did not performed before. Our review article is intended to attract the attention of clinicians to this problem and encourage them to perform such a trial in patients with AMI.

All clinical trials contain information only on the use of GLP1R agonists in patients with type 2 diabetes. Therefore, we think that comparing doses of GLP1R agonists in patients with diabetes and rats with coronary artery occlusion is inappropriate.

Maybe the authors could also include a table about the various compounds or describe some of them in more detail. For example, I did not know either ZP2495 (a peptidic agonist at the GLP1 and glucagon receptor) or DMB (a small molecule allosteric GLP1R agonist).

We added this information on ZP2495.

We would like to note that our article is too big (152 References). The review articles published in Int J Mol Sci usually contain 50 – 100 references. I am afraid that Editorial Office will require reducing the text and the number of references. Reviewer #3 has already requested that the text should be shortened and unnecessary discussion removed. Therefore, we would not like to increase discussion.

The manuscript would benefit from proofreading by a native speaker. I suggest the following improvements (this list is not exhaustive):

We have corrected the manuscript based on the reviewer's recommendation:

line 42: instead of “polypeptide increases …” -> “GLP1 increases …” OK

line 49: instead of “The GLP1R is the G protein-coupled receptor” -> “The GLP1R is a G protein-coupled receptor” OK

line 80: instead of “In 1992, polypeptide exendin-4 was isolated …” -> “In 1992, the polypeptide exendin-4 was isolated …” OK

line 86: instead of “In 1991, it was found the selective peptide GLP1R antagonist.” -> “In 1991, a selective peptide GLP1R antagonist was found.” OK

line 105: instead of “… for example, resistant to enzyme hydrolysis erythropoietin …” -> “… for example erythropoietin …” OK

line 135-136: instead of “it was reported on the neuroprotective effects of liraglutide” -> “neuroprotective effects of liraglutide were reported” OK

line 231: instead of “exendin-4 increased a number of surviving cardiomyocytes” -> “exendin-4 increased the number of surviving cardiomyocytes” OK

line 312: the sentence “H9c2 cells exposed to H/R” seems to be incomplete OK

line 337: instead of “However, investigators used small groups” -> “However, the investigators used small groups” OK

line 516: instead of “Compund mitigated …” -> “DMB mitigated …” OK

line 523-524: instead of “It was used 10 rats in each group. Exendin (25 nmol/kg/day = 105 µg/kg/day intraperitoneally) for 6 weeks.” -> “10 rats in each group were injected intraperitoneally (25 nmol/kg/day = 105 µg/kg/day) with exendin-4 for 6 weeks”

I think it is impossible to inject rats intraperitoneally.

line 631: instead of “Adverse cardiac remodeling can be developed in diabetes.” -> “Adverse cardiac remodeling can develop in diabetes.” OK

line 634: instead of “However, exendin-4 only improved …” -> “However, only exendin-4 improved …” OK

line 777: Instead of “It has been found that following protein kinases are …” -> “It has been found that the following protein kinases are …” OK

line 824: what is “course administration”? for 4 weeks.

We corrected English using your recommendations and vocabulary from native English speakers. PMID: 14506302  PMID: 26426469   PMID: 29882685 PMID: 30215858

Sincerely yours, corresponding author Leonid N Maslov

Reviewer 2 Report

Comments and Suggestions for Authors

The manuscript tried to demonstrate that the activation of the GLP1R promotes cardiac tolerance to I/R. Indeed, GLP1R agonists are cardioprotective when used acutely and chronically before ischemia and when used in reperfusion. 

The reviewer suggest the following extensive major revisions: 

- I suggest to retype the title as follow: "Peptides are cardioprotective drugs of the future: the receptor and signaling mechanisms of the cardioprotective effect of glucagon-like peptide-1". 
- My suggestion in the introduction is to create a simple table of the two peptides, to render easier the text. 
- in paragraph 2, again, I suggest to report the residues of the different types of exeding in a table, to render easier the main text. 
- in vivo and in vitro should be written in italics in all the text. 
The reviewer suggests to shorten all the paragraphs as the text is difficult to read. Please remove the non-necessary information.
- write the meaning of the acronyms in the legend of all the figures. 

Comments on the Quality of English Language

Moderate editing of English language required. Some phrases are very difficult to understand. 

Author Response

Dear colleagues, thank you very much for your recommendations that helped to improve the quality of our article.

The manuscript tried to demonstrate that the activation of the GLP1R promotes cardiac tolerance to I/R. Indeed, GLP1R agonists are cardioprotective when used acutely and chronically before ischemia and when used in reperfusion. 

The reviewer suggest the following extensive major revisions: 

- I suggest to retype the title as follow: "Peptides are cardioprotective drugs of the future: the receptor and signaling mechanisms of the cardioprotective effect of glucagon-like peptide-1".  OK

- My suggestion in the introduction is to create a simple table of the two peptides, to render easier the text. OK

- in paragraph 2, again, I suggest to report the residues of the different types of exeding in a table, to render easier the main text. OK

- in vivo and in vitro should be written in italics in all the text. OK

- The reviewer suggests to shorten all the paragraphs as the text is difficult to read. Please remove the non-necessary information. OK

- write the meaning of the acronyms in the legend of all the figures. OK

Comments on the Quality of English Language

Moderate editing of English language required. Some phrases are very difficult to understand. 

We corrected English using your recommendations and vocabulary from native English speakers. PMID: 14506302  PMID: 26426469   PMID: 29882685 PMID: 30215858

Sincerely yours, corresponding author Leonid N Maslov

Reviewer 3 Report

Comments and Suggestions for Authors

Major Comments:

1.      “Peptides are cardioprotective drugs of the future”-This seems a too generalized statement. The authors may revise the title!

2.      The authors stated about the cardioprotective effect of glucagon-like peptide-1 in the title, however the abstract and the manuscript emphasized the cardioprotective effect of GLP1R agonists!

3.      The authors may discuss about the modulators of glucagon-like peptide-1.

4.      The authors may discuss about the preclinical or clinical trials of glucagon-like peptide-1 if any as a cardioprotective drug.

5.      The authors may discuss about the limitations and potential adverse effects of using glucagon-like peptide-1 as a cardioprotective drug.

Author Response

Dear colleagues, thank you very much for your recommendations that helped to improve the quality of our article.

  1. “Peptides are cardioprotective drugs of the future”-This seems a too generalized statement. The authors may revise the title! The authors stated about the cardioprotective effect of glucagon-like peptide-1 in the title, however the abstract and the manuscript emphasized the cardioprotective effect of GLP1R agonists!

We corrected a title using Your recommendations and recommendations reviewer # 1.

  1. The authors may discuss about the modulators of glucagon-like peptide-1.

We would like to note that our article is too big (152 References). The review articles published in Int J Mol Sci usually contain 50 – 100 references. I am afraid that Editorial Office will require reducing the text and the number of references. Reviewer #1 has already requested that the text should be shortened and unnecessary discussion removed. Therefore, we would not like to increase this discussion.

  1. The authors may discuss about the preclinical or clinical trials of glucagon-like peptide-1 if any as a cardioprotective drug.

We discussed all preclinical studies of GLP1R agonists inducing cardioprotective effect. Clinical trials of GLP1R agonists as cardioprotective drugs did not performed before. Our review article is intended to attract the attention of clinicians to this problem and encourage them to perform such a trial in patients with AMI.

  1. The authors may discuss about the limitations and potential adverse effects of using glucagon-like peptide-1 as a cardioprotective drug.

We added information on the limitation and side effects of GLP1R agonists in clinical studies. Clinical trials of GLP1R agonists as cardioprotective drugs did not perform before.

Sincerely yours, corresponding author Leonid N Maslov

Round 2

Reviewer 2 Report

Comments and Suggestions for Authors

The authors have made revisions to the text in accordance with the reviewer's suggestions. I believe the manuscript is now suitable for publication in its current form.

The English has been improved, and the text is now more comprehensible and fluid.